# The Putative S1PR1 Modulator ACT-209905 Impairs Growth and Migration of Glioblastoma Cells In Vitro

**DOI:** 10.3390/cancers15174273

**Published:** 2023-08-26

**Authors:** Sandra Bien-Möller, Fan Chen, Yong Xiao, Hanjo Köppe, Gabriele Jedlitschky, Ulrike Meyer, Céline Tolksdorf, Markus Grube, Sascha Marx, Mladen V. Tzvetkov, Henry W. S. Schroeder, Bernhard H. Rauch

**Affiliations:** 1Department of General Pharmacology, University Medicine Greifswald, 17475 Greifswald, Germany; sandra.bien@med.uni-greifswald.de (S.B.-M.);; 2Department of Neurosurgery, University Medicine Greifswald, 17475 Greifswald, Germany; 3Division of Pharmacology and Toxicology, School of Medicine and Health Sciences, Carl von Ossietzky, Universität Oldenburg, 26129 Oldenburg, Germany

**Keywords:** glioblastoma, S1P, S1P receptor 1, ACT-209905, THP-1, macrophages

## Abstract

**Simple Summary:**

In this paper, we report on the inhibition of glioblastoma (GBM) cell growth and migration by the putative S1PR1 modulator ACT-2009905, using appropriate in vitro models. This work is based on our previously published finding that S1PR1 expression is strongly up-regulated in human glioblastoma samples and the fact that there is an association between S1PR1 with patients’ survival times. We now show that pharmacological modulation by the putative S1PR1 modulator ACT-209905 inhibits GBM cell growth and migration. Furthermore, we investigated the influence of co-culture of GBM cells with THP-1 cells as a model for human monocytes, showing pro-tumoral effects and arguing for a complex interplay between GBM cells, immune cells and underlying signaling pathways. We believe that this manuscript fits the interests of the readership of the journal *Cancers* as it addresses the impact of alternative therapeutic options using ACT-209905 for improving GBM therapy.

**Abstract:**

Glioblastoma (GBM) is still a deadly tumor due to its highly infiltrative growth behavior and its resistance to therapy. Evidence is accumulating that sphingosine-1-phosphate (S1P) acts as an important tumor-promoting molecule that is involved in the activation of the S1P receptor subtype 1 (S1PR1). Therefore, we investigated the effect of ACT-209905 (a putative S1PR1 modulator) on the growth of human (primary cells, LN-18) and murine (GL261) GBM cells. The viability and migration of GBM cells were both reduced by ACT-209905. Furthermore, co-culture with monocytic THP-1 cells or conditioned medium enhanced the viability and migration of GBM cells, suggesting that THP-1 cells secrete factors which stimulate GBM cell growth. ACT-209905 inhibited the THP-1-induced enhancement of GBM cell growth and migration. Immunoblot analyses showed that ACT-209905 reduced the activation of growth-promoting kinases (p38, AKT1 and ERK1/2), whereas THP-1 cells and conditioned medium caused an activation of these kinases. In addition, ACT-209905 diminished the surface expression of pro-migratory molecules and reduced CD62P-positive GBM cells. In contrast, THP-1 cells increased the ICAM-1 and P-Selectin content of GBM cells which was reversed by ACT-209905. In conclusion, our study suggests the role of S1PR1 signaling in the growth of GBM cells and gives a partial explanation for the pro-tumorigenic effects that macrophages might have on GBM cells.

## 1. Introduction

In contrast to several other cancers, the survival time of patients suffering from glioblastoma (GBM) is still devastating. So far, no groundbreaking improvement in the therapeutic management of GBM has been achieved, and the median survival time is only about 15 months. A complete surgical resection of GBM is nearly impossible due to the highly infiltrative behavior of the tumor, and almost all GBM patients experience a relapse despite surgery and aggressive radiochemotherapy [1]. A versatile signaling lipid, which is intensively discussed to be an important pro-tumoral molecule, is sphingosine-1-phosphate (S1P). S1P is involved in the proliferation, migration and invasion of both healthy and malignant cells [2]. Interestingly, an increase in S1P has been associated with the proliferation and invasion of GBM and other cancers that display a propensity for brain metastasis [3]. In GBM patients’ tissue, S1P levels are highly elevated compared to non-malignant brain tissue samples [4]. Both astrocytes and glioma cells are able to release S1P into the extracellular space [5,6,7]. The upregulation of S1P is discussed as a key strategy to increase the survival of and enhance the resistance properties of GBM cells. S1P is generated intracellularly from sphingosine, which is derived from the hydrolysis of ceramide during the degradation of sphingomyelin and glycosphingolipids, catalyzed by the sphingosine kinases 1 and 2 (SphK1/2) [8]. A poor prognosis in patients with GBM was shown to be correlated with high SphK1 expression [9], and the inhibition of SphK1 results in the growth arrest of GBM cells both in vitro and in vivo [9,10]. The knockdown of SphK2 was described to inhibit GBM growth even more potently than that of SphK1, thus indicating that both SphKs might be involved in GBM cell progression [9].

In addition to its intracellular actions, S1P binds to five G-protein-coupled receptors, referred to as S1P receptors 1–5 (S1PR1-5), in an autocrine and paracrine manner [11,12]. These S1PRs display tissue-specific expression patterns with overlapping functions but also with some opposite effects [13]. In GBM tissue, S1PR1, S1PR2, S1PR3 and S1PR5 are found to be overexpressed, whereas S1PR4 is missing in GBM cells [14,15]. Furthermore, Kaplan–Meier analyses have demonstrated an association between S1PR1 and S1PR2 with patients’ survival times [15]. The stimulation of cultured GBM cells with S1P results in enhanced cell proliferation mediated by S1PR1, S1PR2 and S1PR3, whereas S1PR5 inhibits S1P-stimulated cell proliferation [16,17,18]. Van Brocklyn et al. showed that the motility of GBM cells is stimulated by S1P and involves both S1PR1 and S1PR3 [9]. For S1PR2, both the inhibition and induction of migration and motility were demonstrated [16,17,19], and our own studies argue for a pro-migratory role of S1PR1 and S1PR2 in cultured GBM cells, too [15].

In addition, Fingolimod (FTY720), a functional antagonist of S1PR1, is able to slow the growth of intracranial xenograft tumors in nude mice, augments the therapeutic effect of the cytostatic temozolomide and has a strong antiproliferative impact on GBM cells [20,21].

However, the GBM tissue represents a complex formation of tumor cells itself and diverse non-malignant cells such as endothelial cells, microglia or immunocompetent cells from peripheral blood [22]. Up to 50% of the bulk tumor in GBM is constituted by macrophages, making them the primary innate immune cells in the tumor microenvironment [23,24]. In this regard, accumulating evidence supports that S1P acts as a key signal in the cancer extracellular milieu [3]. S1P creates a tracking gradient for innate and adaptive immune cells and regulates diverse cellular processes that are important for immune responses, including the differentiation and migration of immune cells such as lymphocytes, natural killer cells and macrophages [25]. The increased S1P content within GBM and the diminished circulating S1P level in GBM patients might support monocyte migration from the peripheral blood into the tumor [4,26]. All five S1P receptors are also expressed by microglial cells and macrophages varying according to the activation state of these cells, with S1PR1 and S1PR3 being mostly involved in their migration [27,28,29,30].

However, based on several studies, S1P signaling and, particularly, S1PR1 seem to have an important role in the complex machinery which triggers GBM growth and progress. Thus, the purpose of this study was to discover the antitumoral efficiency of S1PR1 modulation in GBM cells by the agonist ACT-209905, which represents a ponesimod analogue that acts as a functional antagonist with immunomodulating properties, probably mechanistically, in a similar manner as ponesimod [31,32,33]. The impacts of ACT-209905 on the growth and migratory properties of human (LN18, U87MG and primary GBM cells) and mouse (GL261) GBM cells, as well as the potential synergistic effects in combination with temozolomide and other S1PR1/2 antagonists, were determined. Furthermore, the influence of the co-culture of GBM cells with THP-1 cells as a model for human monocytes was analyzed, showing pro-tumoral effects and arguing for the idea that there is a complex interplay between GBM cells, immune cells and underlying signaling pathways.

## 2. Materials and Methods

### 2.1. Cultivation of Adherent GBM Cells

The human glioblastoma cell lines LN-18 and U-87MG (ATCC, Manassas, VA, USA) and the murine glioblastoma cell line GL261 (kindly provided by M. Synowitz, University Medicine Kiel, Germany, originally from National Cancer Institute, MD, USA) were used. Further, primary glioblastoma (prGBM) cells from a surgically resected patient’s sample were isolated as described by [34]. The cultivation of GBM cells was carried out at 37 °C, 95% relative humidity and 5% carbon dioxide fumigation in DMEM supplemented with 10% FCS, 2 mM glutamine and non-essential amino acids. By employing a PCR-based assay, all cell lines were routinely monitored for potential mycoplasma contamination.

Cells were stimulated with the noncommercial S1PR1 modulator ACT-209905 provided by Actelion Pharmaceuticals (Basel, Switzerland) [31,32,33]. The S1PR1 antagonist W146 and the S1PR2 antagonist JTE-013 were both from Tocris Bioscience (Bristol, UK). Sphingosine-1-Phosphate and temozolomide were purchased from Sigma-Aldrich (Deisenhofen, Germany).

### 2.2. Cultivation of GBM Cells as Stem-Like Neurospheres

LN-18 neurospheres, which are thought to be enriched in cancer stem cells [35], were cultured using the NeuroCult™ NS-A Proliferation Kit (Human, STEMCELL Technologies, Cologne, Germany), with 20 ng/mL rh EGF, 10 ng/mL rh bFGF and 0.0002% heparin added according to the manufacturer’s protocol.

Briefly, after trypsinization of LN-18 cells, the cell suspension was centrifuged at 1000 rpm for three minutes. After aspirating the supernatant, cells were washed twice with PBS, followed by centrifugation at 1000 rpm for three minutes. Then, cells were resuspended in NeuroCult™ NS-A Proliferation medium and cultured for seven days, followed by the application of ACT-209905 at different time points. In parallel, adherent LN-18 cells were cultured for the same time period and in the same conditions but using the conventional DMEM medium (with 10% FCS, 2 mM glutamine, and non-essential amino acids) for comparison.

### 2.3. Co-Culture of GBM Cells with the Human Monocytic Cell Line THP-1

The monocytic suspension cell line THP-1 was purchased from ATCC (Manassas, VA, USA) and cultured at 37 °C in a 5% CO_2_ and 95% humidified air incubator in DMEM supplemented with 10% FCS, 2 mM glutamine and 2 mM non-essential amino acids. For co-cultivation experiments, GBM cells were inoculated either in 96-well plates (cell viability and vitality analyses) or 12-well plates (migration analyses), and after 24 h, the medium was removed, and the cells were cultured with fresh medium containing different numbers of THP-1 cells for a further 72 h, as indicated in the respective figure legends. After 72 h, the medium containing THP-1 cells was removed, and the GBM cells were washed at least three times with PBS until no THP-1 cell residues were seen under the microscope. Afterwards, cell viability with resazurin assay, cell vitality with crystal-violet staining or migration analyses were performed as described below in the respective sections. To use the THP-1-conditioned medium, THP-1 cells (0/mL, 7500/mL, 15,000/mL, 30,000/mL, 45,000/mL and 60,000/mL, respectively) were cultured in 6-well plates in serum-starved medium for 24 h. Then, THP-1 cells were centrifuged at 800 rpm for 5 min, and the supernatant was transferred to a new sterile tube and stored at −20 °C until use.

### 2.4. Cell Viability (RESAZURIN Assay) and Cell Vitality (Crystal-Violet Staining) Analyses

GBM cells were inoculated in 96-well plates, with 10,000 cells in each well. After 24 h, the medium was removed, and the cells were then cultured for different time points, using fresh medium containing S1P or the respective S1P receptor agonist/antagonists. Then, the medium was removed and replaced with fresh medium containing 10% resazurin (PromoCell, Heidelberg, Germany), and the 96-well plates were then returned to the 37 °C cell incubator for 1.5 h to 2 h. Fluorescence readings were recorded using a multiplate reader (Tecan Infinite M200, Crailsheim, Germany; excitation wavelength of 530 nm and emission wavelength of 590 nm). Data were calculated as the percentage of cell viability of solvent-treated cells.

Afterwards, the supernatant was aspirated, and cells were rinsed once with PBS. Then, cells were fixed for 10 min with 4% paraformaldehyde, followed by gently rinsing three times with PBS. A total of 50 μL of 0.5% crystal-violet staining solution (Sigma-Aldrich Co., Deisenhofen, Germany) was added to each well, and the plates were incubated for 10 min at room temperature. Then, the cells were washed with A. dest. several times until the dye stopped coming off. After washing, the plate was gently tapped on filter paper to remove any remaining liquid. Finally, the stained cells were treated with 100 µL SDS solution (1%) and incubated for 5 min, shaking gently on a rocking shaker to dissolve the coloring. The optical density was determined to be 560 nm (OD560) with a multiplate reader (Tecan Infinite M200, Crailsheim, Germany). All treated samples were normalized to the untreated control cells (100%).

### 2.5. Scratch-Wound-Healing Assay

A wound-healing assay was performed to investigate the undirected cellular migration. GBM cells were inoculated into a 24-well plate at a density of 150,000 cells per well. Once the cells had built a complete confluent monolayer, a scratch (“wound”) was made with a 100 µL pipette tip into the cell layer. Afterwards, the culture medium was removed, and cells were then washed twice with PBS to remove detached cells and prevent their reattaching. Then, prewarmed culture medium containing 0.5% FCS and 5 mM hydroxyurea as a proliferation inhibitor was added. Wounds were imaged using the PALM Robo software of the AxioVision HXP 120C microscope (Carl Zeiss Microscopy, Jena, Germany), and the exact location of the images was saved to analyze the same area after incubation with the test compounds. Following pre-incubation with hydroxyurea (1 mM) for one hour, cells were treated with ACT-209905 and/or S1P at different concentrations in hydroxyurea-containing medium. Then, images of the scratch were taken again, and the wound width was calculated (Software AxioVision SE64 Rel. 4.9, Carl Zeiss Microscopy).

### 2.6. Boyden Chamber Assay

We performed chemotaxis assays in transwell Boyden chambers, with membranes possessing a pore diameter of 8 µm and being precoated with collagen for 24 h at room temperature. After incubation, membranes were washed in PBS, hung up to dry and stored until use at 4 °C. On the day of the chemotaxis analysis, the migration stimuli were placed in the lower chamber, and the membrane and the upper chamber were fixed onto it. Then, 50 µL of cell suspension was added to each well of the upper chamber (5 × 10^3^ cells in 0.5% FCS medium). The cells were then allowed to migrate towards the lower Boyden chambers, which contained the stimuli, for 3 h at 37 °C. DMEM containing 10% FCS was used a positive control for migration. Afterwards, cells on the upper side of the membrane were scraped, the membrane was washed in PBS and the cells on the lower side of the membrane were fixed in 4% paraformaldehyde for 5 min. Then, the membrane was stained in crystal violet solution for 1 min (crystal violet solution: 5.4 mL ethanol [99.8%], 1.5 mL crystal violet solution, Sigma Aldrich Co., Deisenhofen, Germany). The membrane was washed twice in A. dest. and hung up to dry. The following day, the membrane was stuck between a microscopy slide and a cover glass with Entellan^®^ (Merck Kg, Darmstadt, Germany), followed by a drying period of about 24 h. Images of the membrane were made using the PalmRobo microscope, and stained cells were counted using Image J software (IJ 1.46r, National Institutes of Health, USA). All treated samples were normalized to the untreated control cells (100%).

### 2.7. Caspase 3 Activity Assay

Caspase 3 activity was measured using a commercially available kit (BioVision Inc., Milpitas, CA, USA). Cells were seeded in 6-well plates at a density of 200,000 cells/well. After two days, cells were incubated with the respective test compounds for 48 h. After treatment, the supernatant, including the detached cells due cell death, was collected in a conical tube. Adherent cells were scraped and collected in the same tube and centrifuged at 250× *g* for 10 min. Afterwards, cells were rinsed with PBS two times and centrifuged again at 250× *g* for 10 min. The supernatant was discarded, while the cell pellet was lysed by the addition of 60 µL Lysis Buffer (provided in the kit) and incubated on ice for 10 min, followed by centrifugation at 10,000× *g* for 1 min. Then, 50 μL of cell lysate and 50 μL of 2X Reaction Buffer 3 were mixed per well of a 96-well-plate. Prior to using the 2X Reaction Buffer 3, 10 μL of DTT stock (provided by the kit) per 1 mL was added. To each reaction well, 5 μL of Caspase 3 substrate (DEVD-AFC, 1 mM) was added, and the plate was incubated at 37 °C for 1–2 h, as suggested by the supplier. Then, plates were read on a microplate reader (Infinite M200 Microplate reader, TECAN, Crailsheim, Germany), using an excitation wavelength of 400 nm and an emission wavelength of 505 nm. All treated samples were normalized to the untreated control cells (100%).

### 2.8. Real Time RT PCR

Total RNA was isolated using PeqGold RNAPure (PeqLab, Erlangen, Germany) and reversely transcribed using the High-Capacity cDNA Reverse Transcription Kit (Applied Biosystems by Life Technologies, Weiterstadt, Germany). The following Gene Expression Assays on Demand (Applied Biosystems) were used: S1PR1, Hs00173499_m1; S1PR2, Hs01015603_s1; S1PR3, Hs01019574_m1; S1PR4, Hs02330084_s1; S1PR5, Hs009 24881_m1; endogenous control eukaryotic 18S rRNA, Hs03003631_g1; GAPDH, Hs02758991_g1; and ß-Actin, 4310881E-0910026. Quantitative Real-Time PCR was performed on a 7900 HT Fast Real-Time PCR system and QuantStudio 7 Flex Real-Time PCR System from Applied Biosystems. The mRNA content of target genes (S1PR) was normalized to the mean of 18S rRNA, β-Actin and GAPDH and is expressed relative to the control cells (100%).

### 2.9. Western Blotting

GBM cells were scraped or trypsinized, transferred to a 1.5 mL tube and centrifuged at 10,000 rpm for 3 min. Afterwards, the cell pellets were resuspended in lysis buffer (50 mM Tris-HCl, 100 mM NaCl, 0.1% Triton X-100 and 5 mM EDTA, pH 7.4) supplemented with 1 mM leupeptin, 1 mM aprotinin and 250 μg/mL sodium vanadate and incubated on ice for 30 min, followed by centrifugation at 12,000 rpm for 5 min at 4 °C. The supernatant was used for protein determination, using the BCA method. Subsequently, after denaturation in Laemmli SDS sample buffer at 95 °C for 5 min, 30–50 μg of protein was separated on 10% SDS polyacrylamide gels. The tank blot system (Bio-Rad, Hempstead, UK) was used for the immunoblotting of the separated proteins to Whatman^®^ nitrocellulose membranes, which were afterwards blocked in 5% FCS in Tris-buffered saline containing 0.05% Tween 20 (TBST) for 1 h at room temperature. The membrane was incubated overnight at 4 °C, under rotating conditions, with the following primary antibodies diluted in TBST: mouse anti-CD133 (Merck Millipore, Darmstadt, Germany; Novus Biologicals, Cambridge, UK; Cell Signaling Technology, Boston, USA); anti-Nestin (STEMCELL Technologies Inc., Vancouver, BC, Canada); mouse anti-GFAP (Cell Signaling Technology, Boston, MA, USA); rabbit anti-S1PR1 (Abcam, Eugene, OR, USA); mouse anti-PCNA (Santa Cruz Biotechnology, Inc., Heidelberg, Germany); anti-phospho-p38 and anti-total-p38 (both Cell Signaling Technology, Boston, MA, USA); rabbit anti-phospho-AKT and rabbit anti-total-AKT1 (both Cell Signaling Technology, Boston, MA, USA); mouse anti-phospho-ERK1/2 and mouse anti-total-ERK1/2 (both Cell Signaling Technology, Boston, MA, USA); and mouse anti-GAPDH (Meridian Life Science Inc., Memphis, TN, USA; Bio-Rad Laboratories GmbH, München, Germany). Afterwards, the membrane was rinsed three times with TBST, for five minutes each time, followed by incubation with the secondary fluorescence-labeled antibody for one hour at room temperature, with gentle shaking: anti-mouse IRDye^®^ 680 or 800 CW and anti-rabbit IRDye^®^ 680 or 800 CW (all from Li-cor Bioscience GmbH, Bad Homburg, Germany). After incubation, the membrane was rinsed with TBST at room temperature on a horizontal shaker three times, for five minutes each time. Fluorescence signals were detected using the Odyssey^®^ CLx Imaging System (Li-cor Bioscience GmbH, Bad Homburg, Germany). The relative fluorescence intensities of the specific bands were calculated and normalized to GAPDH as a loading control.

### 2.10. Flow Cytometry

For quantitative evaluation of surface molecules on GBM cells, flow cytometry was used. After the treatment of cells with the respective compounds or co-culture with THP-1, the cells were harvested and washed once with ice-cold PBS (pH 7.4), followed by centrifugation at 5000 rpm for 5 min. Then, the supernatant was carefully removed, and the cells were fixed for 10 min in 4% paraformaldehyde and then washed again with PBS. Afterwards, fixed cells were incubated in a total volume of 20 µL for 30 min at room temperature, in the dark, with one of the following antibodies: APC anti-CD11b (Biolegend, San Diego, CA, USA), PE anti-CD54/ICAM-1 (IOTest, Beckman Coulter, Krefeld, Germany), FITC anti-CD62p (BD Pharmingen, Heidelberg, Germany), PE-Vio770 anti-CD97 (Miltenyi Biotech, Teterow, Germany), APC anti-CD142 (eBioscience, Thermo Fisher Scientific, Waltham, MA, USA), APC anti-CD163 (Miltenyi Biotec, Teterow, Germany), PE anti-CD166/ALCAM (IOTest, Beckman Coulter, Krefeld, Germany) or APC anti-CD206 (Biolegend, San Diego, CA, USA). The respective isotype antibodies were used as negative controls: APC mouse IgG1 (Biolegend, San Diego, CA, USA), PE-Vio770 mouse IgG1 and APC mouse IgG1 (both Miltenyi Biotec, Teterow, Germany), FITC mouse IgG1 (BD Pharmingen, Heidelberg, Germany), APC mouse IgG1 (eBioscience, Thermo Fisher Scientific, Waltham, MA, USA) and PE mouse IgG1 (IOTest, Beckman Coulter, Krefeld, Germany). Then, the cells were resuspended in 500 µL PBS for flow cytometric analysis. For each sample, 5000 cells were analyzed. Sample acquisition was performed at a Guava easyCyte 8 (Merck Millipore, Darmstadt, Germany), using GuavaSoft™ 2.7 Software. Typically, events were first gated on the basis of shape and granularity (FSC-SSC). For a comparison of target surface expression on THP-1 and GBM cells, the percentage of stained cells was used, and the median fluorescence intensity (MFI) relative to isotype control was calculated according to the following formula: median fluorescence of stained target sample—median fluorescence of isotype control.

### 2.11. Immunofluorescence Staining of S1PR1

LN-18 cells were seeded on coverslips and incubated for 48 h with 20 µM of ACT or the respective solvent control (methanol). Cells were fixed with 1% paraformaldehyde in phosphate buffer saline (PBS) (30 min at room temperature (RT)), washed and blocked with 20% FCS, 1% saponin (in PBS). Incubation with a polyclonal rabbit anti-S1PR1 antibody (Abcam, Cambridge, UK; dilution 1:50 in PBS supplemented with 10% FCS and 1% saponin) was performed overnight at 4 °C. After being washed with PBS, the coverslips were incubated for 1 h (RT) with anti-rabbit secondary antibodies conjugated with either Alexa Fluor^®^-488 or Alexa Fluor-568^®^ (Thermo Fisher Scientific, Waltham, MA, USA). For staining control, incubations with only the secondary antibodies were also performed. Co-staining of the cytoskeletal marker F-actin was performed with Alexa Fluor^®^-568-labeled phalloidin (Thermo Fisher Scientific, Waltham, MA, USA), which was added to the secondary antibody incubations (1 h, RT). After being washed with PBS, the coverslips were embedded using a mounting medium with 4′,6-diamidino-2-phenylindole (DAPI) as a counterstain for nuclei (Roti^®^-Mount FluorCare DAPI, Carl Roth GmbH, Karlsruhe, Germany). Fluorescence micrographs were taken using the confocal laser scanning microscope LSM780 (Zeiss, Jena, Germany).

### 2.12. Statistical Analysis

Statistical analyses were performed using the GraphPad Prism software 5.0. (GraphPad Software, Inc., San Diego, CA, USA). All data were presented as mean ± SD of at least three independent experiments. Pairwise comparisons were performed using Student’s *t*-test and the Mann–Whitney U test, as indicated. More than two groups were compared by one-way ANOVA and Dunnett’s multiple comparison test or Bonferroni post-hoc test. Non-linear regression modeling (log(inhibitor) vs. normalized response—variable slope) was used to derive a dose–response curve and IC50 values. Statistical significances were defined as * or # *p* < 0.05; ** or ## *p* < 0.01; and *** or ### *p* < 0.001.

## 3. Results

### 3.1. Expression of S1PR1 in GBM Cells

First, we investigated the expression of S1PR1 in human GBM cells by immunoblotting. As shown in Figure 1a, both the human LN-18 and U-87MG cell lines, as well as the murine GL261 cell line, expressed S1PR1 at a similar level. Furthermore, S1PR1 expression was evident in the human peripheral blood monocytic cell line THP-1, which was used for co-culture experiments. In addition, S1PR1 was detectable in primary GBM cells (prGBM) that were isolated from a patient suffering from this brain tumor. These prGBM cells also expressed the astrocytic marker GFAP, as well as CD133 and Nestin (Figure 1b); both of them are discussed as GBM stem cell marker, arguing for their suitability as a GBM cell model. Furthermore, the prGBM cells responded to S1P stimulation with an increased cell viability, as also seen for the murine GL261 cells (Figure 1c).

It is well known that S1PR1 modulators such as fingolimod and ponesimod cause the degradation of S1PR1 after its internalization [36,37]. As shown in Appendix A, the treatment of LN-18 GBM cells with ACT-209905 also resulted in a strongly reduced S1PR1 protein expression, as determined by immunoblot analysis (Appendix A, 72 h, 10 µM of ACT209905) and immunofluorescence staining in most of the cells (Appendix A, 48 h, 20 µM of ACT-209905). In line with the results for the LN-18 cell viability, 5 µM of AC-20995 had no effect on S1PR1 expression in the immunoblot analyses.

### 3.2. Influence of S1PR1 Modulator ACT-209905 on GBM Cell Viability and Vitality

To analyze whether the S1PR1 modulator ACT-209905 influences the growth behavior of GBM cells in vitro, we determined the viability (resazurin assay) and vitality (crystal-violet staining) of both human (LN-18, U-87MG) and mouse (GL251) GBM cell lines, as well as primary human GBM cells. As shown in Figure 1d–f and in the Appendix A, treatment of GBM cells with different ACT-209905 concentrations (0.5 to 50 μM) for 48 h and 72 h resulted in significant changes in cell viability and vitality.

In human LN-18 GBM cells, viability was significantly attenuated by 10, 20 and 50 μM of ACT-209905 after 48 h to 36.2%, 18.5% and 1.43%, respectively (Figure 1d, upper panel). LN-18 cell viability showed a similar decrease after 72 h treatment with 10, 20 or 50 µM of ACT-209905 to 20.8%, 15.7% and 0.37%, respectively (Appendix A, upper panel). The crystal-violet assays for LN-18 cells (Figure 1d, lower panel) showed that cell vitality after the application of 10, 20 or 50 µM of ACT-209905 was significantly reduced to 60.7%, 43.7% and 31.0% after 48 h and to 41.9%, 33.0% and 24.6% after 72 h, respectively (Appendix A, lower panel). In comparison to LN-18 cells, the response of human U-87MG cells was somewhat less to 10 µM of ACT-209905, but cell viability was also strongly diminished to 13.2% and 0.02% after 72 h of treatment with 20 and 50 µM of ACT-209905 (Appendix A, upper panel). The crystal-violet assays further showed that the vitality of U-87MG cells was decreased to 30.3% and 21.2% after 72 h of treatment with 20 and 50 µM of ACT-209905 (Appendix A, lower panel).

In murine GL261 cells, already 1 µM of ACT-209905 resulted in a significant decrease in cell viability to 72% after 48 h, and this was potentiated by increasing the ACT-209905 concentration, resulting in a maximal reduction of 7.3% at 50 μM of ACT-209905 (Figure 1e, upper panel). Seventy-two hours after the application of ACT-209905, the viability of the GL261 cells was reduced to 65.9%, 33.8%, 19.7% and 13.5% at 5, 10, 20 and 50 µM, respectively (Appendix A, upper panel). Again, crystal-violet assays (Figure 1e, lower panel) confirmed the growth-inhibiting effect of ACT-209905 in GL261 cells, showing that the vitality was significantly decreased to 62.4%, 45.7% and 39.1% after 48 h of treatment with 10, 20 or 50 µM of ACT-209905, respectively. Seventy-two hours after application, the cell vitality was similarly affected by ACT-209905 (Appendix A, lower panel), but, paradoxically, low ACT-209905 concentrations (0.5 and 1 µM) increased GL261’s vitality from 130 to 135%. This growth-promoting phenomenon of ACT-209905 in murine GL261 cells was also slightly observed in human LN-18 but not in U-87MG or prGBM cells (see above).

In addition to the well-established GBM cell lines (LN-18, U-87MG and GL261), we analyzed the influence of ACT-209905 on prGBM cells. As demonstrated in Figure 1f (upper panel), the viability of prGBM cells was significantly reduced after 48 h to 80.4%, 67.0%, 47.7%, 21.9% and 8.2% at 1, 5, 10, 20 and 50 µM of ACT-209905, respectively. Again, the crystal-violet assays showed similar results for the vitality of prGBM cells, with diminished values of 82.4%, 55.9%, 25.0% and 10.7% at 5, 10, 20 and 50 µM of ACT-209905, respectively (Figure 1f, lower panel). Consistently, the concentration-dependent effect of ACT-209905 in reducing the viability and vitality of prGBM cells was potentiated after 72 h of treatment (Appendix A).

Additionally, the IC50 values of ACT-209905 for all investigated GBM cells were calculated and are summarized in Table 1.

### 3.3. Combination of ACT-209905 with W146 (S1PR1 Antagonist) and JTE-013 (S1PR2 Antagonist) Enhances the Growth Inhibitory Effects

Next, we investigated whether an enhancement in the growth-inhibitory effect of ACT-209905 could be reached through a combination with other S1PR1 (W146) or S1PR2 (JTE-013) antagonists, using a low cytotoxic concentration of ACT-209905, as well as of W146 and JTE-013 (5 µM each). As shown in Figure 1g,h, as well as Appendix A, in comparison with W146 and JTE-013, the treatment with ACT-209905 caused the strongest reduction in GL261 cell viability to a value of 73.4%. The dual application of the S1PR antagonists mentioned above resulted, in any case, in a significantly potentiated loss of GL261 cell viability, i.e., to 38.9% (ACT-209905 + W146), 60.4% (ACT-209905 + JTE-013) and 56.7% (W146 + JTE-013), respectively (Figure 1g, upper panel). In general, the crystal-violet assays showed similar results for the GL261 cell vitality, but the effect of the used substances was somewhat less pronounced (Figure 1g, lower panel). Similar effects of the ACT-209905 application alone and in combination with W146 or JTE-013 were observed in prGBM cells. The single application of ACT-209905 reduced the viability of prGBM cells to 69.2%, whereas the combination with W146 caused a decrease to 49.3%, and with JTE-013, there was a decrease to 63.0%, respectively. As seen for GL261, the vitality of prGBM cells was affected in the same way. The data obtained for human LN-18 and U-87MG GBM cells are presented in Appendix A, showing the enhanced effect of the combined application of ACT-209905 with JTE-013 compared to the single administration of ACT-209905, too.

### 3.4. Combination of ACT-209905 with the Chemotherapeutic Agent Temozolomide Showed Synergistic Effects

The standard chemotherapeutic drug for patients suffering from GBM is the alkylating agent temozolomide. Therefore, we next analyzed whether the combined application of ACT-209905 with temozolomide has synergistic effects regarding the cytotoxic efficiency (Figure 1i,j and Appendix A). In GL261 cells, the administration of 100 µM temozolomide alone caused a decrease in the viability to 56.6%, similar to the single application of 10 µM of ACT-209905, with a value of 57.7%; moreover, the dual treatment with both compounds resulted in a significantly stronger effect, with a reduction to 35.3% (Figure 1i). This synergistic effect was also seen for 20 µM of ACT-209905 (28.1%, single application) and its combination with temozolomide (15.2%). The crystal-violet assays confirmed the enhancing effects of the dual administration of ACT-209905 and temozolomide in GL261 cells (Appendix A). In Figure 1j and Appendix A, the data for the prGBM cells are shown which demonstrate nearly the same impact of single and dual treatments with ACT-209905 and temozolomide on cell viability and vitality. The single application of temozolomide caused a decrease in cell viability to 41.4%. The application of 10 or 20 µM of ACT-209905 alone reduced the viability to 55.1% and 26.5%, whereas the combination with temozolomide led to viability values of 14.6% (together with 10 µM of ACT-209905) and 3.17% (together with 20 µM of ACT-209905). In addition, the results obtained for human LN-18 and U-87MG GBM cells affirmed the intensified cytotoxic effect of a combined application of ACT-209905 and temozolomide (Appendix A).

### 3.5. Activation of Caspase 3 by ACT-209905 in GBM Cells

To determine if apoptosis might be induced, we measured caspase 3 activity after the treatment of GBM cells for 48 h with ACT-209905. Figure 2a represents the data obtained for the cell lines GL261, LN-18 and U-87MG. Whereas the application of 5 µM of ACT-209905 did not cause an increase in caspase 3 activity, the treatment with 20 µM of the compound doubled caspase 3 activity, and 50 µM of ACT-209905 caused a much stronger caspase 3 activation of up to 486.6% (LN-18), 245.9% (GL261) and 206.0% (U-87MG), respectively.

PrGBM cells also respond to ACT-209905 with a strong induction of caspase 3 activity to 395.1% (20 µM) and 660.1% (50 µM). The simultaneous administration of 2.5 µM of S1P together with ACT-209905 slightly reduced the activation of caspase 3, but this was not statistically significant (Figure 2b).

### 3.6. Impact of ACT-209905 on Viability of Stem-Like GBM Neurospheres

Since former studies from our group have demonstrated that LN-18 cells represent a suitable cell model to investigate the behavior of GSCs as stem-like GBM neurospheres [38], we further investigated whether ACT-209905 treatment is also able to diminish the viability of LN-18 stem-like cells. As seen in Figure 2c, the application of 10 µM of ACT-209905 caused a much stronger cytotoxic effect in LN-18 neurospheres compared to the adherent counterpart at the time points analyzed. ACT-209905-treated adherent LN-18 cells showed a viability of 84.1%, 59.0% and 26.7% after 24 h, 48 h and 72 h, whereas LN-18 neurospheres had values of 36.6%, 4.10% and 1.14% at the same time points, respectively. To further characterize the role of S1PRs in GSC, we measured their mRNA expression patterns. S1PR1 expression was increased by about 5-fold in LN-18 neurospheres compared to their adherent counterparts (Appendix A).

### 3.7. ACT-209905 Negatively Influences Migration of GBM Cells

Furthermore, the impact of ACT-209905 on GBM cell migration was analyzed with the scratch-wound-healing assay (after 16 h; Figure 2d,e). As seen in Figure 2d,e, stimulation with 2.5 µM of S1P doubled the migration rate of prGBM cells. This pro-migratory influence of S1P was inhibited by pretreatment with ACT-209905 from a wound closure of 21.0% to 13.2% (1 µM), 9.9% (10 µM) and 6.12% (20 µM) which was already below the basal migration capacity (7.52%) of control cells (Figure 2d). The single application of 10 and 20 µM of ACT-209905 also reduced the migration of prGBM cells, but this effect was not statistically significant when compared to control cells treated with solvent only.

The application of another S1PR1 antagonist, called W146, showed a similar reduction in the S1P-induced migration of prGBM cells, as shown in Figure 2e. The migration rate was diminished from 22.1% wound closure (2.5 µM of S1P alone) to 18.7% (1 µM of W146 + S1P), 14.4% (5 µM of W146 + S1P) and 10.6% (10 µM of W146 + S1P), respectively. Again, a slight decrease in wound closure was observed when W146 was added alone, but this was not statistically significant as well.

### 3.8. Co-Cultivation with Monocytic THP-1 Cells Increases GBM Cell Growth

Since it is known that immune cells can invade tumor tissue, influencing tumor growth behavior, we investigated whether the co-cultivation of GBM cells with THP-1 monocytic cells has any impact on the viability and vitality of the brain tumor cells. Figure 3a–d and Appendix A show the results for prGBM cells and the murine GL261 cell line. For both the prGBM and GL261 cells, the co-cultivation with an increasing number of THP-1 cells resulted in enhanced viability and vitality. Maximal effects were seen at a THP-1 cell concentration of 3.0 to 4.5 × 10^4^/mL, with an increased viability level of 240.7% and 272.3% for prGBM cells (Figure 3a), as well as 168.8% and 179.3% for GL261 cells (Figure 3c). Consistently, the vitality values were elevated to 168.0% and 171.9% for the co-cultivation of prGBM cells with 3.0 and 4.5 × 10^4^/mL THP-1 cells (Appendix A), as well as to 139.6% and 155.8% for the co-cultivation of GL261 cells with 3.0 and 4.5 × 10^4^/mL THP-1 cells (Appendix A), respectively.

A similar viability- and vitality-enhancing effect was observed for the cultivation of GBM cells with THP-1-conditioned media (Figure 3b,d and Appendix A). The viability and vitality were significantly increased to 234.8% and 159.8% when prGBM cells were cultured with THP-1-conditioned media (3.0 × 10^4^/mL; Figure 3b and Appendix A), as well as to 179.8 and 143.5% for the cultivation of GL261 with THP-1-conditioned media (3.0 × 10^4^/mL; Figure 3d and Appendix A). Studies on human LN-18 GBM cells revealed the same pro-tumorigenic effect of co-cultivation with conditioned media of THP-1 cells with values of 265.4% and 193.5% (3.0 × 10^4^/mL, Appendix A), respectively.

### 3.9. ACT-209905 Inhibits the Pro-Survival Impact of THP-1 Cells on GBM Cells

Next, we investigated whether the application of ACT-209905 is able to reverse the growth-promoting effect of THP-1 co-cultivation on GBM cells (Figure 3e–h and Appendix A). The co-cultivation with THP-1 cells (3.0 × 10^4^/mL) together with the application of 10 µM of ACT-209905 inhibited the increase in viability, decreasing it from 244.3% to 124.3% for prGBM cells (Figure 3e) and from 134.3% to 120.5% for GL261 cells (Figure 3g). Again, the co-cultivation of GBM cells with THP-1-conditioned media (from 3.0 × 10^4^/mL cells) together with the application of ACT-209905 showed the same effect, with a reduction in cell viability from 166.9% to 92.9% for prGBM cells (Figure 3f) and from 126.0% to 97.7% for GL261 cells (Figure 3h).

For the human LN-18 cells, the co-cultivation with THP-1 cells also resulted in a significant increase in viability to 170.9% (1.5 × 10^4^/mL) and 236.8% (3.0 × 10^4^/mL), and these values were reduced by the treatment with ACT-209905 to 82.6% and 128.9%, respectively (Appendix A). Once again, a similar effect was seen for the cultivation of LN-18 cells with THP-1 cells after the application of ACT-209905, with a reduction from 124.6% and 173.1% (only THP- cells; 1.5 or 3.0 × 10^4^/mL) to 61.8% and 98.5%, respectively (Appendix A).

### 3.10. Co-Cultivation with Monocytic THP-1 Cells Increases GBM Migration, which Is then Inhibited by ACT-209905

The co-cultivation with THP-1 cells significantly stimulated the migration of prGBM cells (Figure 4a) from a wound-closure value of 9.76% (without THP-1) to 15.2% (1.5 × 10^4^/mL THP-1) and 23.9% (3.0 × 10^4^/mL THP-1). A similar increase in migration was seen when prGBM cells were cultured with THP-1-conditioned media (Figure 4b) from 7.60% (without THP-1) to 17.8% (1.5 × 10^4^/mL THP-1) and 23.9% (3.0 × 10^4^/mL THP-1), respectively.

Treatment with 10 µM of ACT-209905 significantly decreased the pro-stimulatory impact of THP-1 cells on GBM cell migration (Figure 4c–f and Appendix A) from a wound closure of 24.5% (3.0 × 10^4^/mL THP-1 only) to 16.9% in prGBM cells (Figure 4c) and from 30.7% (3.0 × 10^4^/mL THP-1 only) to 17.5% in mouse GL261 cells, respectively (Figure 4d).

Figure 4e,f represents the data obtained from the wound-closure assay for the co-culture of GBM cells with THP-1-conditioned media. In prGBM cells, wound closure was increased from 14.5% (control) to 21.4% and to 29.1% by conditioned medium of 1.5 × 10^4^/mL and 3.0 × 10^4^/mL THP-1 cells, and these values were which after the additional application of 10 µM of ACT-209905 to 14.5% and 15.5%, respectively. The determination of the directed migration using the Boyden Chamber confirmed these results (Figure 5a). The cultivation of prGBM cells in THP-1-conditioned media significantly enhanced chemotaxis to 136.3% (1.5 × 10^4^/mL) and 149.3% (3.0 × 10^4^/mL), but it was then inhibited by the administration of 10 µM of ACT-209905, decreasing to 102.8% and 113.1%, respectively.

The same effect was seen in mouse GL261 cells with an enhanced wound closure upon culture in THP-1-conditioned media, increasing from 9.40% (control) to 14.1% and 25.9% (1.5 × 10^4^/mL and 3.0 × 10^4^/mL); again, this was decreased to 9.04% and 15.2%, respectively. Again, using Boyden Chamber to measure the direct cell migration, a similar pro-migratory effect of THP-1-conditioned media was observed (Figure 5c), with a significant elevation of GL261 chemotaxis to 135.7% and 154.3% (1.5 × 10^4^/mL or 3.0 × 10^4^/mL THP-1 cells) which was reduced to 101.9% and 114.6%, respectively.

The commercial S1PR1 antagonist W146 was also investigated in regard to its influence on THP-1 effects (Appendix A). In human prGBM cells, the stimulating effect of THP-1 cells of conditioned media was significantly diminished from 32.0% (3.0 × 10^4^/mL THP-1 cells) and 27.2% (conditioned media of 3.0 × 10^4^/mL THP-1 cells) to 17.9% and 19.3% after the application of 10 µM of W146, respectively (Appendix A). Directed cell migration determined by Boyden Chamber Assays showed a similar anti-migratory effect of W146, as it reduced the significantly enhanced chemotaxis of prGBM cells from 133.4% and 158.8% (only conditioned media of 1.5 × 10^4^/mL or 3.0 × 10^4^/mL THP-1 cells) to 106.6% and 132.5% upon its administration, respectively (Figure 5b).

The data obtained for mouse GL261 cells showed nearly the same effect as prGBM cells. The wound closure, which was increased by the co-culture with THP-1 cells or conditioned media, was reduced from 29.3% (3.0 × 10^4^/mL THP-1 cells) and 27.5% (conditioned media of 3.0 × 10^4^/mL THP-1 cells) to values of 14.1% and 17.9% by W146, respectively (Appendix A). In Figure 5d, the directed chemotaxis using Boyden Chambers is shown, demonstrating that the elevated GL261 migration via cultivation in THP-1-conditioned media was decreased from 135.7% and 158.9% (only conditioned media of 1.5 × 10^4^/mL or 3.0 × 10^4^/mL THP-1 cells) to 100.2% and 115.7% by W146, respectively.

### 3.11. Potential Relationship between Signaling Pathways and ACT-209905 Effects and THP-1-Induced Proliferation and Migration of GBM Cells

To identify potential signaling pathways modulated by the co-cultivation of GBM cells with THP-1 cells and treatment with ACT-209905, immunoblotting analyses were performed. As seen in Figure 6, we analyzed PCNA (proliferating cell nuclear antigen) as the key regulator for cell-cycle progression and, thus, a proliferation marker, as well as the kinases p38, AKT1 and ERK1/2, which are known to be activated by S1P receptors such as S1PR1.

A single application of ACT-209905 slightly decreased PCNA protein expression in both prGBM and mouse GL261 cells. The incubation of human prGBM cells with 3.0 × 10^4^/mL THP-1 cells or conditioned media significantly enhanced PCNA expression to 230.6% and 197.9%, which were reduced to 173.1% and 156.8% by ACT-209905, respectively (Figure 6a,b). A similar increase in PCNA expression upon cultivation with 3.0 × 10^4^/mL THP-1 cells or conditioned media with values of 246.9% and 212.9% was observed for GL261 cells, and this was reverted to 179.0% and 137.7% through the administration of 10 µM of ACT-209905, respectively (Figure 6c,d).

The phosphorylation of p38 (P-p38) was significantly enhanced to 210.6% and 200.6% by 3.0 × 10^4^/mL of THP-1 cells or conditioned media in prGBM cells (Figure 6a,b), and this increase in P-p38 was reversed by 10 µM of ACT-209905 (174.5 and 173.1%). Figure 6c,d show the phosphorylation status of p38 in GL261 cells with increased levels of 175.4% (3.0 × 10^4^/mL THP-1 cells only) and 188.3% (conditioned media), which were significantly reduced to 141.9% and 130.8% by 10 µM of ACT-209905, respectively.

As seen in Figure 6a,b, the AKT1 phosphorylation (P-AKT1) of prGBM cells was enhanced when the cells were co-cultured with 3.0 × 10^4^/mL of THP-1 cells or conditioned media to values of 221.1% and 185.3%, and the application of 10 µM of ACT-209905 resulted in reductions to 166.5% and 150.1%, respectively. Mouse GL261 GBM cells showed a similar elevation in AKT1 phosphorylation to 216.1% and 190.9% for co-cultivation with 3.0 × 10^4^/mL THP-1 cells or conditioned media, and these values were also decreased to 136.9% and 146.1% by ACT-209905, respectively (Figure 6c,d).

Finally, the phosphorylation of ERK1/2 (P-ERK1/2) was determined to show enhanced ERK1/2 activation (196.2% and 163.1%) when the prGBM cells were co-cultured with 3.0 × 10^4^/mL of THP-1 cells or conditioned media, and the values were reduced to 140.6% and 127.1% by ACT-209905 (Figure 6a,b). Again, the same effect was seen for GL261 cells with increased ERK1/2 phosphorylation values of 205.8% (3.0 × 10^4^/mL THP-1 cells only) and 198.5% (THP-1-conditioned media), which were then reduced to 142.6% and 148.5% by ACT-209905, respectively (Figure 6c,d).

The single application of 10 µM of ACT-209905 caused a slightly reduced phosphorylation of all analyzed kinases, but this was not significant at the investigated conditions.

### 3.12. M2 Marker Surface Expression on THP-1 Cells and Modulation of Pro-Migratory Surface Molecules by ACT-209905 and THP-1 in GBM Cells

Firstly, the surface expression of the general macrophage marker CD11b, as well as of CD163 and CD204, two commonly accepted specific markers for M2-type macrophage, was evaluated using FACS analyses. As shown in Figure 7a, 22.8% of THP-1 cells were positive for CD11b, with an MFI of 15.6. Surprisingly, both ACT-209905 (10 µM) and S1P (2.5 µM) caused a significant more-than-two-fold increase in the percentage of CD11b-positive cells, as well as an about-three-fold-elevated median fluorescence intensity (MFI) for CD11b on the surface of THP-1 cells (Figure 7a). The M2 macrophage marker CD163 was detectable on about 45% of THP-1 cells (about 30 MFI), and its expression was not affected by ACT-209905 or S1P (Figure 7b). CD206, another M2 macrophage marker, was also expressed by about 20% of THP-1 cells, and while it was reduced by ACT-209905 to 8.82%, it was increased by S1P to 30.2% (Figure 7c). Concordantly, the MFI of CD206 was significantly diminished from 12.7 (control) to 5.61 by ACT-20995, but it was then elevated to 23.5 by S1P. Since our study shows an alteration in GBM cell migration by the S1PR1 modulator ACT-209905 and co-culture with THP-1 cells or conditioned media, we examined if surface molecules with migratory functions might also be influenced. It was seen that only about 7.22% of prGBM cells expressed CD54 (ICAM1) under control conditions but ACT-209905 slightly increased the percentage of CD54-positive cells to about 10% (Figure 7d, middle). More important, the MFI of CD54 surface expression was strongly decreased by 10 µM of ACT-209905 from 16.7 (control) to 0.81 (Figure 7d, right panel), whereas the co-culture with THP-1 cells or THP-1-conditioned medium (CM) increased the MFI of CD54 to 27.6 and 20.9, respectively, which was clearly prevented by ACT-209905 (MFI of CD54: 8.71 and 3.59). For CD62P (P-Selectin), we found a reduced percentage of positive cells after the treatment of prGBM cells with ACT-209905 from 18.1% (control) to 13.1%, but the MFI of CD62P was not significantly changed under all tested conditions (Figure 7e). In contrast, the co-culture with THP-1 cells increased the percentage of CD62P positive prGBM cells to 23.6%, which was then significantly reduced by ACT-209905 to 14.9%. For CD166 (ALCAM), the percentage of positive prGBM cells was only very slightly diminished by ACT-209905 under all conditions, in the same way, and the MFI of CD166 was also strongly reduced by ACT-209905 to about half that of the control (Figure 7f). The co-culture with THP-1 cells or conditioned medium had no impact on the CD166 surface expression of prGBM cells. Furthermore, CD97 and CD142 (Tissue Factor), which were both described to play a role in the invasive phenotype of GBM cells [39,40], were unchanged in prGBM cells.

## 4. Discussion

To date, there is no curative or considerably life-prolonging therapy for patients suffering from GBM. Thus, the search for alternative, maybe-targeted therapeutic options is urgently needed. The lipid mediator S1P influences numerous cellular processes, such as proliferation, survival, migration and angiogenesis [41]. All of these processes have been shown to be important for progression of the highly aggressive GBM and thus emerging S1P-targeting treatment as a promising therapy for GBM. However, a more thorough understanding of S1P signaling will help to provide more information for therapeutic drug design.

To date, five different mammalian receptor subtypes for S1P (S1PR1-5) with different expression patterns and partly opposing cellular effects were identified. With the exception of S1PR4, which is missing in GBM, all S1PR subtypes are found to be overexpressed in GBM [14]. S1PR1, S1PR2 and S1PR3 were shown to be elevated in patient brain-tumor samples compared to normal brain samples, and Kaplan–Meier analyses have demonstrated an association between S1PR1 and S1PR2 with patients’ survival times [15]. The functional S1PR1 antagonist FTY720 (fingolimod) slows the growth of intracranial xenograft tumors in mice [20], enhances the cytotoxic effect of temozolomide and has strong antiproliferative effects on GBM cells used in clinical trials for different cancers, including GBM (Clinicaltrials.gov, NCT02490930). The final results of that study are not yet available.

Our study also helps highlight the role of S1PR1 in the proliferation and migration of human and mouse GBM cells, since the modulation of this receptor subtype by the novel compound ACT-209905 blocks both processes. ACT-209905, a chiral amino pyridine derivative of ponesimod, is an S1PR1 modulator with immunomodulating properties that could be of use in treating autoimmune diseases [31,32,33]. In all investigated GBM cell lines (human and murine), as well as prGBM cells, ACT-209905 had a dose-dependent cytotoxic effect. The IC50 values for ACT-209905 after 72 h of treatment are between 5 and 30 µM depending on the GBM cells used; thus, these values lie in a similar range as that observed for FTY720 (4.6 to 25.2 µM) in GBM cells [21]. The cytotoxic effect of ACT-209905 was confirmed by the activation of caspase 3 after 48 h, arguing for the induction of apoptosis by the compound.

It has been demonstrated that different S1P receptor subtypes exert opposite effects in GBM cell migration. S1PR1 and S1PR3 mediate migratory responses and amplify those exerted by other growth factors, whereas S1PR2 inhibits cell migration [17,42]. Besides the inhibitory effect of ACT-209905 on cell viability, in our present study, the migratory and invasive capacity of GBM cells was also reduced by this S1PR1 modulator, particularly when cells were stimulated with S1P as a pro-migratory molecule. The inhibition of migration was seen as a response to a short treatment with ACT-209905 (e.g., 3 h to 24 h), where no or only very slight cytotoxic effects of the compound were visible. Another S1PR1 antagonist, called W146, caused similar effects on GBM cell proliferation and migration as ACT-209905 did in our study. Synergistic effects were observed when ACT-209905 and W146 or JTE-013, a S1PR2 antagonist, were combined. Even more relevant, the dual application of ACT-209905 and temozolomide also resulted in much stronger cytotoxicity compared to the single treatment. Altogether, these data support the potential role of S1PR1 in both the proliferation and migration of GBM cells, making this protein an excellent target for therapeutic options. Apart from this, the GBM stem cell (GSC) phenotype and survival seem to involve S1P signaling [41,42,43,44]. GSCs have been found to be the cells with the highest rate of S1P synthesis and secretion in GBM, and GSCs have been reported to be more responsive to extracellular S1P than their GBM parent cells, possibly due to their higher S1PR1, -2 and -4 expression. Thus, we were excited to see what happens when stem-like GBM neurospheres are treated with ACT-209905. Interestingly, the cytotoxic effect of ACT-209905 was even more potent in LN18 neurospheres compared to its adherent counterpart. These data suggest that ACT-209905 damages both normal and stem-like GBM cells. This finding is in agreement with the abovementioned observation of an elevated S1P synthesis and secretion rate and an increased expression of S1PR1 in GSC, as observed in this study and in the study by Annabi and colleagues [45].

However, the GBM microenvironment contains several non-neoplastic cells, including infiltrating and resident immune cells, and S1P is discussed as a key signal in the reciprocal crosstalk between tumor cells and recruited normal cells which seems to be crucial for tumor growth and progression [46]. Within GBM, the S1P content is elevated [4], whereas a diminished circulating S1P level in GBM patients was described [26] which might support immune cell migration from the peripheral blood into the tumor. Tumor-associated macrophages (TAMs) are the dominant non-tumoral population accounting for 30–40% of the cells in the tumor and are inversely correlated with patient survival [47,48]. These macrophages have two main polarization states: classically activated M1 and alternatively activated M2 subtypes [49,50]. Intratumoral TAMs are believed to mainly have an anti-inflammatory M2 phenotype that contributes to tumor growth and migration rather than exerting antitumor protection [51,52], and a poor prognosis for GBM patients was associated with the abundance of M2 macrophages [52]. It has been further shown that S1P signaling polarizes macrophages to the M2 phenotype [53], and S1PR1 has been linked to macrophage activation [54]. Our co-cultivation experiments with the monocytic cell line THP-1 or its conditioned medium showed a strong induction of viability and migration of GBM cells. THP-1 is a human leukemic monocyte cell line that is thought to be able to exist in both monocyte- and macrophage-like states [55]. Of note, in other studies, THP-1 monocytes were first differentiated from macrophages by using PMA (Phorbol 12-myristate 13-acetate), followed by stimulation with IL-4 and IL-13 in order to obtain M2-polarized macrophages or with IFN-gamma and LPS for classical macrophage activation (M1) [56]. In our experiments, we used undifferentiated THP-1 cells, which grow as suspension culture, and THP-1-conditioned media, and both conditions are able to increase the viability and migration of GBM cells. These observations clearly suggest that THP-1 cells might have a pro-tumoral M2 phenotype already, without any specific cytokine stimulation (e.g., IL-4 and IL-13) for differentiation. The macrophage phenotype of THP-1 cells was confirmed by the FACS measurement of CD11b, and, more important, the M2 macrophage markers CD163 and CD206 were also present on the surface of THP-1 cells. Interestingly, the expression of CD206 was significantly decreased by ACT-209905, but it was increased by S1P itself, underlying the role of S1P signaling in the pro-tumoral M2 phenotype. Of note, in our study, THP-1 cells were cultured in a similar medium as GBM cells that contained non-dialyzed FCS, which, thus, also included hormones and cytokines for THP-1 cell stimulation. Our data indicate that cultured THP-1 cells might secrete factors which mediate the pro-tumorigenic effects through the activation of pro-survival pathways but without needing a direct cell–cell interaction, since the conditioned medium of THP-1 was also sufficient to induce comparable pro-tumoral effects. It is well known that TAMs express and secrete several factors to promote tumorigenesis. Potential candidates might be TGF-β1, which was shown to be secreted by THP-1 cells and to trigger the migration of GBM cells [57]; and IL-10, which is secreted by THP-1 M2 macrophages and promotes tumorigenesis in GBM cells [58]. The precise role of these molecules in THP-1 modulated growth and migration of GBM cells has to be examined in further studies.

Interestingly, our viability and migration analyses showed that the pro-tumoral effect of THP-1 on GBM cells was not detectable when tumor cells were treated with the S1PR1 modulator ACT-209905 in parallel. This could mean two things: either THP-1-induced growth is inhibited due to the cytotoxic effect of ACT-209905 in GBM cells or due to the activation of signaling pathways stimulated by secreted molecules from THP-1 cells by the compound. A cytotoxic effect of ACT-209905 on THP-1 cells seems not to be the key event since the inhibitory effect was also seen when the GBM cells were co-cultured with THP-1-conditioned medium without any cells. To clarify the mediators of the THP-1-induced migration of prGBM cells, we analyzed a panel of pro-migratory surface molecules. Whereas CD97 and CD142 (Tissue Factor), for which a pro-invasive phenotype of GBM cells was already shown [37,38], were not regulated by either ACT-209905 or THP-1 cells, the surface expressions of CD54 (ICAM1), CD62P (P-Selectin) and CD166 (ALCAM) were significantly regulated by ACT-209905, thus arguing for their role in the migration of GBM cells. Of note, only CD54 and CD62P were upregulated by THP-1 co-culture, which was prevented by concomitantly applied ACT-209905, suggesting that these two molecules are important in the THP-1-induced stimulation of GBM cell migration. Particularly for CD62P and CD166, a potential role in the pathogenesis of GBM was described [59,60].

However, it is known that S1P induces AKT1 activation, most probably through a Gi signaling pathway, which signals to a variety of key downstream molecules to suppress cell death and promote cell survival [61]. The S1PR1 itself is understood to couple exclusively to Gi protein, resulting in a cAMP reduction and the activation of the Ras, MAPK, PI3K, AKT1 and PLC pathways [62]. In GBM cells, the survival-promoting effect of S1P has been related to the activation of different pathways, such as the PI3K/AKT1 signaling pathway, which is well known to be involved in the pathogenesis and resistance of GBM [63,64]. Furthermore, the activation of both AKT1 and ERK1/2 in GBM cells after co-culture with THP-1 M2 macrophages has been described [58]. In our study, AKT1 phosphorylation was also significantly enhanced through the co-cultivation of GBM cells with THP-1, which was partly diminished by ACT-209905. This is in accordance with the observation that FTY720, another S1PR1 modulator, reduces the migration and chemotaxis of GBM cells by inhibiting the PI3K/AKT pathway, too [65]. In addition to AKT1 activation, ERK1/2 and p38 were also activated by THP-1 co-cultivation, and, again, a partial reversing effect was observed to be caused by ACT-209905. A reduction in ERK1/2 activation was already described after the treatment of GBM cells with FTY720, another S1PR1 antagonist [20]. Furthermore, the S1P-mediated inhibition of apoptosis appears to occur mainly through Gi-dependent activation of PI3K/Akt signaling, with ERK1/2 and p38 kinase also being involved. Interestingly, p38 kinase was shown to have both positive and negative implications for FTY720-induced apoptosis, suggesting cell-dependent differences. Altogether, these data argue for a complex network between various signaling pathways triggering GBM growth and migration.

## 5. Conclusions

Taken together, our study suggests that S1PR1 plays a role in the growth and progression of GBM, advances our understanding of the complex mechanisms of S1P-mediated signaling in GBM cells, and provides a partial explanation for the pro-tumorigenic effects that macrophages may have on GBM cells, combined with potential underlying mechanisms. Therefore, this study argues for a further preclinical and clinical evaluation of a pharmacological modulation of S1PR1 as a new or complementary therapeutic principle in GBM therapy.

## Figures and Tables

**Figure 1 cancers-15-04273-f001:**
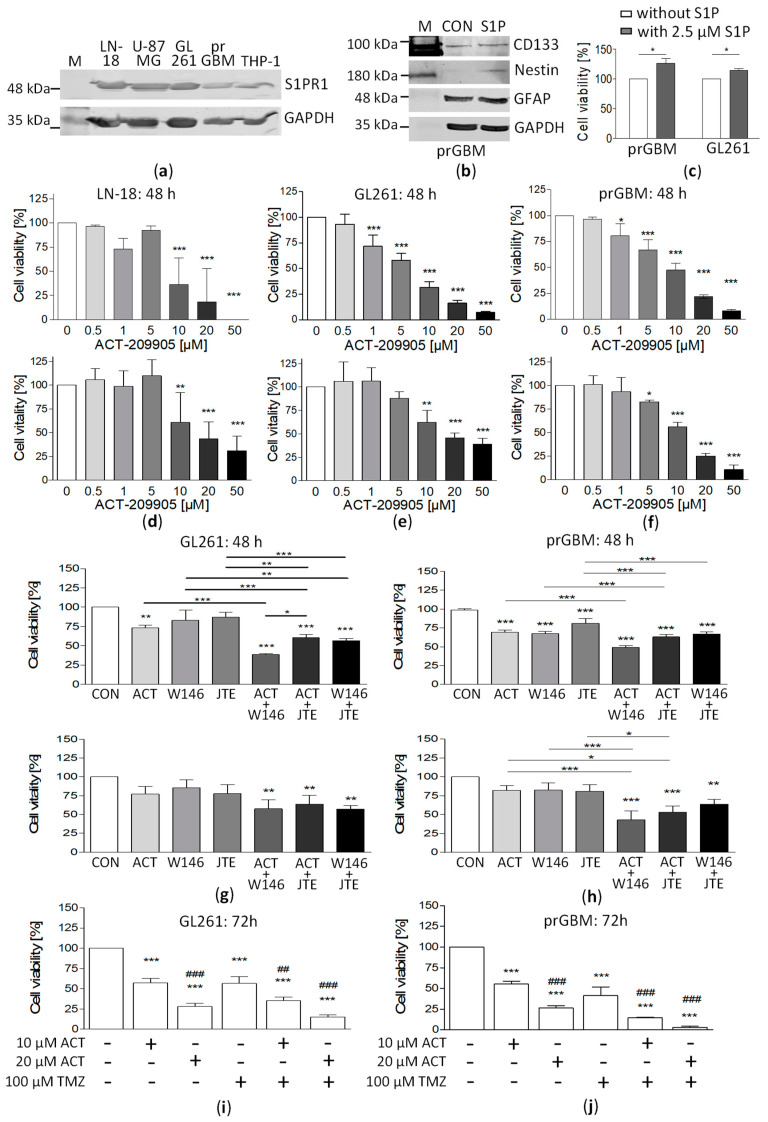
Analysis of S1PR1 expression in GBM cells, as well as the influence of S1PR1 modulator ACT-209905 alone or in combination with other compounds on cell viability of GBM cells. (**a**) Immunoblot analysis of S1PR1 protein expression in murine GL261 cells, in human LN-18 and U-87MG GBM cells, in primary GBM cells (prGBM) isolated from tumor samples and in THP-1 cells. Detection of GAPDH served as internal loading control. A representative blot is displayed. (**b**) Immunoblot analysis of GBM stem cell marker proteins (CD133, Nestin), as well as the astrocytic marker GFAP, in lysates of primary GBM cells (prGBM). Detection of GAPDH was used as loading control. The uncropped blots are shown in File S1. (**c**) Viability of human prGBM and mouse GL261 cells after stimulation with 2.5 µM of S1P for 72 h, as determined by resazurin assay. Cell viability was related to control treated cells (100%, only solvent), mean values and SD, *n* = 3; Mann–Whitney U test, * *p* < 0.05. (**d**–**f**) Determination of cell viability with resazurin assay (upper panel) and cell vitality by crystal-violet staining (lower panel) for LN18 (**d**), GL261 (**e**) and prGBM (**f**) cells after treatment with ACT-209905 (0.5, 1, 5, 10, 20 and 50 μM) for 48 h. (**g**,**h**) Cell viability (resazurin assay, upper panel) and vitality (crystal-violet staining, lower panel) of GL261 (**g**) and prGBM (**h**) cells after application of ACT-209905 (10 µM), W146 (10 µM) and JTE-013 (10 µM) alone or in combination after 48 h. (**i**,**j**) Cell viability determined with resazurin assay after treatment of GL261 (**i**) and prGBM (**j**) cells with ACT-209905 (10 µM or 20 µM) alone or in combination with temozolomide (TMZ, 100 µM) after 72 h. Cell viability or vitality (**d**–**j**) was related to control treated cells (100%, only solvent), mean values and SD, *n* = 3–5, one-way analysis of variance with Dunnett’s multiple comparison test or Bonferroni post hoc test; * *p* < 0.05, ** *p* < 0.01 and *** *p* < 0.001 vs. control (only solvent); ## *p* < 0.01 and ### *p* < 0.001 vs. 10 μM of ACT-209905 alone.

**Figure 2 cancers-15-04273-f002:**
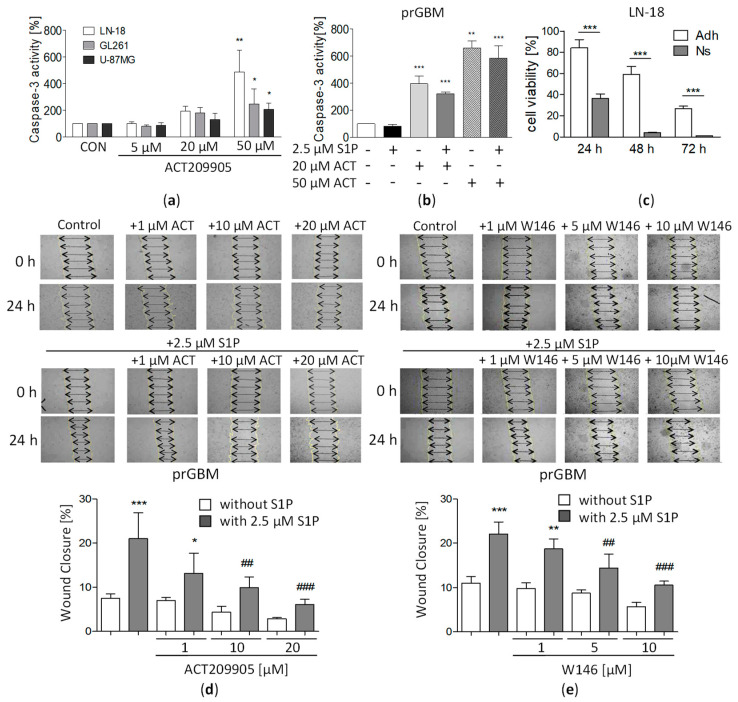
Influence of ACT-209905 on caspase 3 activity, the viability of stem-like neurospheres and migration of GBM cells. Caspase 3 activity of LN-18, GL261, U-87MG (**a**) and prGBM (**b**) cells treated with ACT-209905 (5, 20 and 50 µM) for 48 h. Caspase 3 activity was determined using a commercially available kit. (**c**) LN-18 cells were cultured as adherent cells or stem-cell-like neurospheres and treated with ACT-209905 for 24 h, 48 h and 72 h, followed by analyses of cell viability by using the resazurin assay. (**d**,**e**) Undirected migration of prGBM cells analyzed using the wound-closure assay by setting up a scratch into the cell layer and measuring the wound width at the beginning of the experiment and 24 h after treatment with ACT-209905 ((**d**) 1, 10 and 20 µM) and W146 ((**e**) 1, 5 and 10 µM). The arrows indicate the exact width of the wound. (**a**–**c**) Control cells treated only with the solvent were set to 100%, mean values and SD, *n* = 3–4, one-way analysis of variance with Dunnett’s multiple comparison test or Bonferroni post-hoc test; * *p* < 0.05, ** *p* < 0.01 and *** *p* < 0.001 vs. control; ## *p* < 0.01 and ### *p* < 0.001 vs. 2.5 μM of S1P alone.

**Figure 3 cancers-15-04273-f003:**
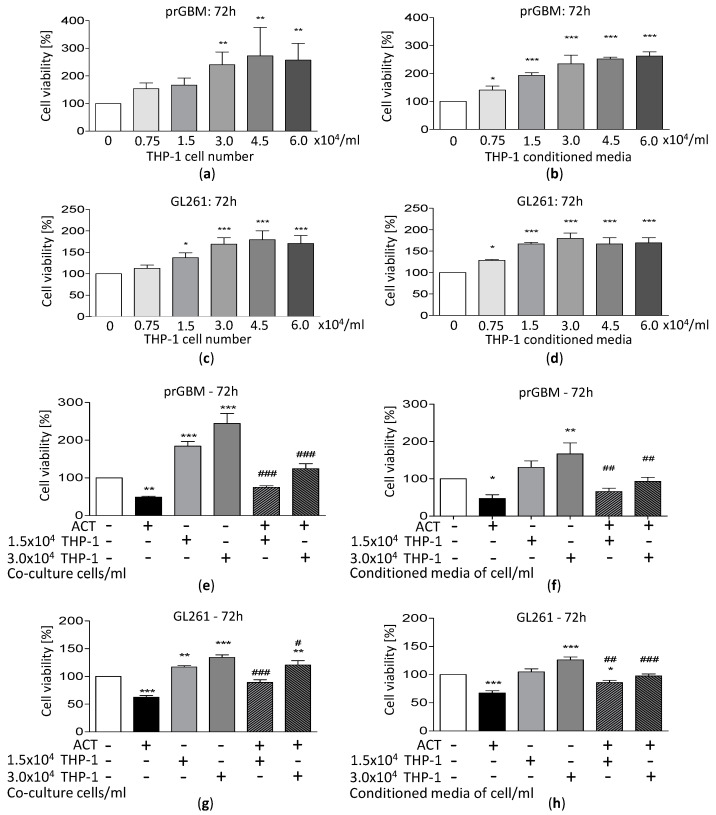
Co-culture effect of THP-1 cells or conditioned media on viability and migration of GBM cells. GBM cells were co-cultured with THP-1 cells at the indicated cell densities or THP-1-conditioned medium for 72 h, and the viability of the cells was measured by the resazurin assay. (**a**,**b**) prGBM cells co-cultured with THP-1 cells (**a**) or THP-1-conditioned medium (**b**) and (**c**,**d**) GL261 cells co-cultured with THP-1 cells (**c**) or THP-1-conditioned medium (**d**) for 72 h; the viability of the cells was measured by the resazurin assay. (**e**,**f**) Co-cultivation of prGBM cells with THP-1 cells (**e**) or THP-1-conditioned medium (**f**) in the presence of ACT-209905 (10 µM), and (**g**,**h**) co-cultivation of GL261 cells with THP-1 cells (**g**) or THP-1-conditioned medium (**h**) in the presence of ACT-209905 (10 µM). (**a**–**h**) Cell viability was related to control cells (100%), mean values and SD, *n* = 3–4, one-way analysis of variance with Dunnett’s multiple comparison test or Bonferroni post hoc test; * *p* < 0.05, ** *p* < 0.01 and *** *p* < 0.001 vs. control; # *p* < 0.05, ## *p* < 0.01 and ### *p* < 0.001 vs. 10 μM of ACT-209905 alone.

**Figure 4 cancers-15-04273-f004:**
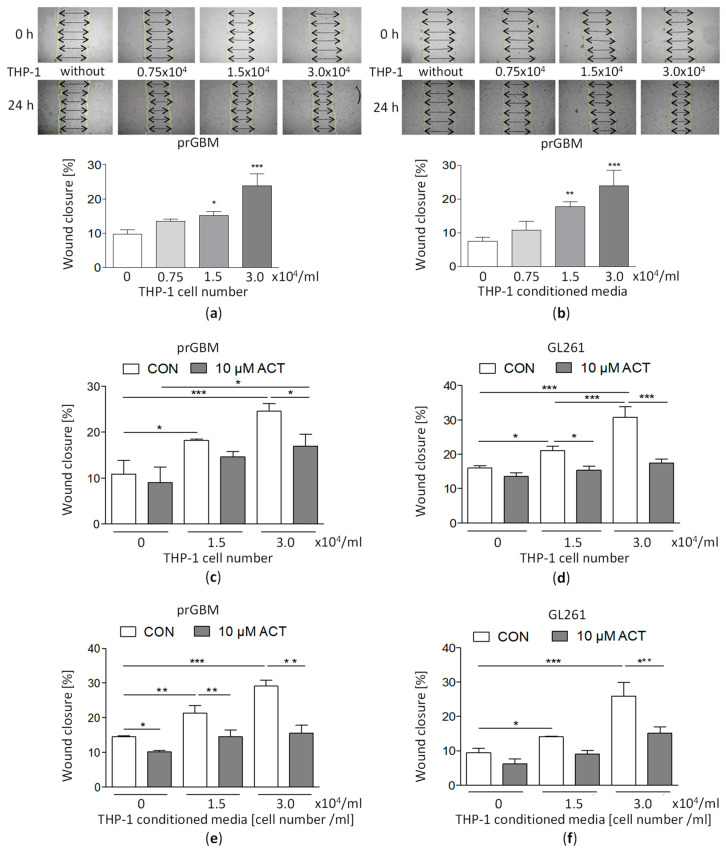
Impact of THP-1 cells or THP-1-conditioned medium together with ACT-209905 application on migration of GBM cells. (**a**,**b**) Migration of prGBM cells analyzed using a wound-closure assay by introducing a scratch into the cell layer and measuring the wound width at the beginning of the experiment and after 24 h of co-culture with THP-1 cells (**a**) or THP-1-conditioned medium (**b**). Representative microscopy images of the scratch area are shown. Mean values and SD, *n* = 3, one-way analysis of variance with Dunnett’s multiple comparison test or Bonferroni post hoc test; * *p* < 0.05, ** *p* < 0.01 and *** *p* < 0.001 vs. control (**c**,**d**) Co-culture of prGBM cells (**c**) or murine GL261 cells (**d**) with THP-1 cells alone or together with application of ACT-209905 (10 µM) and analyses of undirected migration, using a wound-closure assay by introducing a scratch into the cell layer and measuring the wound width at the beginning of the experiment and after 24 h of co-culture. (**e**,**f**) Co-culture of prGBM cells (**e**) or murine GL261 cells (**f**) with THP-1-conditioned media alone or together with application of ACT-209905 (10 µM) and analyses of undirected migration, using a wound-closure assay by introducing a scratch into the cell layer and measuring the wound width at the beginning of the experiment and after 24 h of co-culture. Representative microscopy images of the scratch area are shown in Appendix A. Mean values and SD, *n* = 3, one-way analysis of variance with Bonferroni post hoc test; * *p* < 0.05, ** *p* < 0.01 and *** *p* < 0.001 vs. control.

**Figure 5 cancers-15-04273-f005:**
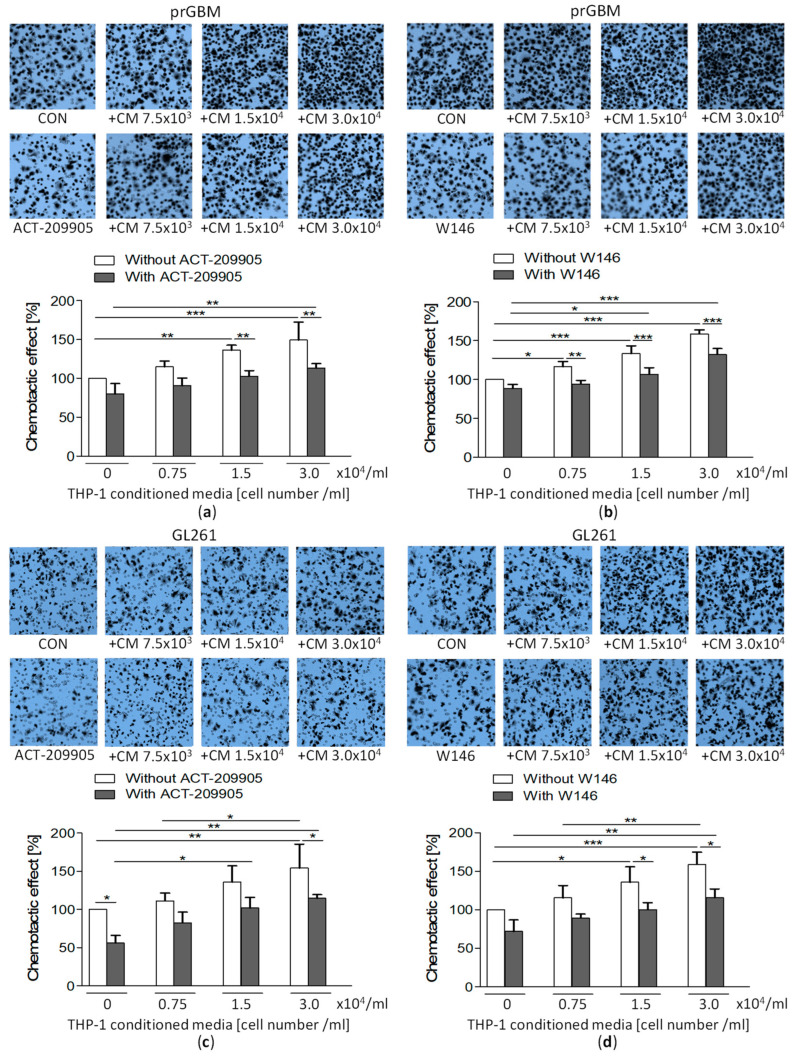
Impact of THP-1-conditioned medium alone or together with ACT-209905 or W146 on directed migration of GBM cells. (**a**,**b**) Co-culture of prGBM cells with THP-1-conditioned medium alone or together with the application of ACT-209905 ((**a**), 10 µM) or W146 ((**b**), 10 µM) and analyses of directed cell migration, using the Boyden chamber assay. (**c**,**d**) Co-culture of murine GL261 cells with THP-1-conditioned medium alone or together with application of ACT-209905 ((**c**), 10 µM) or W146 ((**d**), 10 µM) and analyses of directed cell migration, using the Boyden chamber assay. (**a**–**d**) Cell migration analyses were performed for 3 h; afterwards, cells were fixed on the lower side of the membrane, stained with crystal violet and counted. Representative microscopy images of the fixed and stained cells are shown (Magnification ×40). Counted cells are related to control condition (100%, only solvent), mean values and SD, *n* = 4, one-way analysis of variance with Bonferroni post hoc test; * *p* < 0.05, ** *p* < 0.01 and *** *p* < 0.001 vs. control.

**Figure 6 cancers-15-04273-f006:**
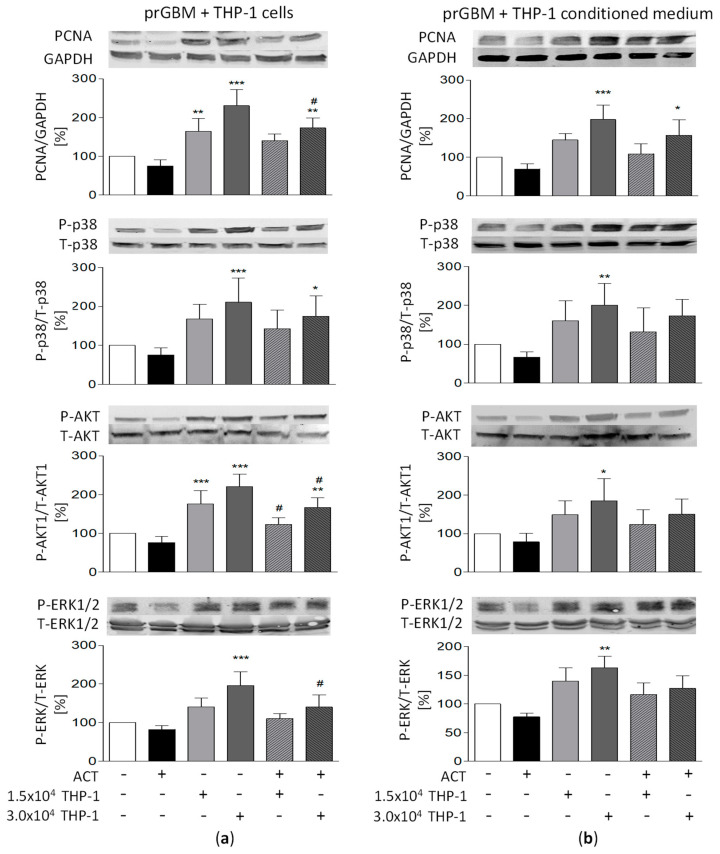
Potential relationship between signaling pathways and THP-1-induced proliferation and migration of human prGBM (**a**,**b**) and murine GL261 cells (**c**,**d**). Immunoblot analysis of PCNA and phosphorylated and total signaling molecules: p38, AKT1 and ERK1/2. GBM cells were co-cultured with THP-1 cells or THP-1-conditioned medium alone or in combination with ACT-209905 (10 µM) for 24 h. Representative blots of at least four independent experiments. The phosphorylated protein level (P) was normalized to the total protein level (T) in the densitometric analysis of each independent experiment; PCNA expression was normalized to GAPDH as loading control. The uncropped blots are shown in File S1. Mean values and SD, *n* = 4–5, one-way analysis of variance with Bonferroni post hoc test; * *p* < 0.05, ** *p* < 0.01 and *** *p* < 0.001 vs. control; # *p* < 0.05, ## *p* < 0.01 and ### *p* < 0.001 vs. the respective ACT-209905 data.

**Figure 7 cancers-15-04273-f007:**
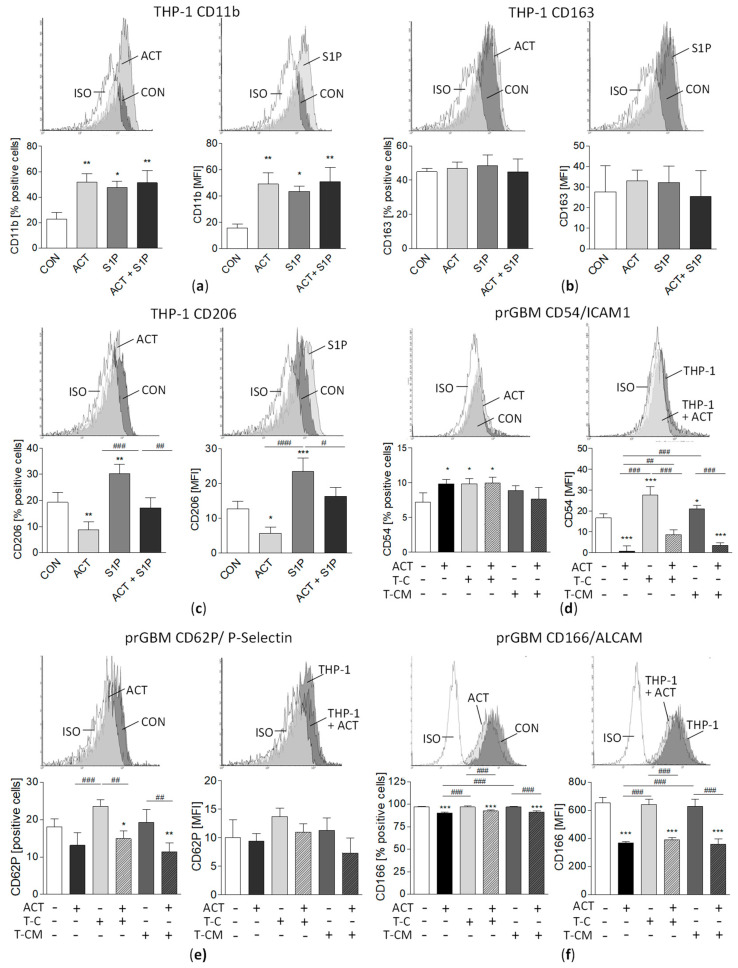
Characterization of macrophage marker surface expression on THP-1, as well as of adhesion and pro-migratory molecules in prGBM cells. Cells were harvested after 24 h of treatment and analyzed for target surface expression, using flow cytometry. The left panels show representative flow cytometry histograms, the middle panels present the percentage of positive cells and the right panels demonstrate the median fluorescence intensity (MFI). The THP-1 cell surface expression of (**a**) CD11b (general macrophage marker), (**b**) CD163 and (**c**) CD204 (M2 marker). THP-1 cells were incubated with ACT-209905 (10 µM) and S1P (2.5 µM) alone or in combination for 24 h, followed by flow cytometry. prGBM cell surface expression of (**d**) CD54 (ICAM1), (**e**) CD62P (P-Selectin) and (**f**) CD166 (ALCAM). GBM cells were co-cultured with THP-1 cells (T-C) or THP-1-conditioned medium (T-CM) either alone or in combination with ACT-209905 (10 µM, ACT) for 24 h, followed by flow cytometry. Data are shown as mean + SD of three-to-four independent experiments. One-way analysis of variance with Bonferroni post hoc test; * *p* < 0.05, ** *p* < 0.01 and *** *p* < 0.001 vs. control; # *p* < 0.05, ## *p* < 0.01 and ### *p* < 0.001 vs. the respective ACT-209905 data.

**Table 1 cancers-15-04273-t001:** IC50 values for ACT-209905 in all investigated GBM cells. Upper panel, non-linear regression with (log(inhibitor) vs. response); lower panel, log(inhibitor) vs. normalized response—variable slope.

	GL261	LN18	U87MG	prGBM
	IC50 (µM)	IC50 (µM)	IC50 (µM)	IC50 (µM)
Viability at 48 h	6.87	19.78	22.53	12.3
Vitality at 48 h	9.32	19.07	40.79	16.16
Viability at 72 h	6.53	16.17	22.52	9.925
Vitality at 72 h	6.15	11.04	29.59	12.39
Viability at 48 h	4.79	9.06	11.54	7.94
Vitality at 48 h	22.12	19.91	17.31	11.42
Viability at 72 h	6.76	9.36	9.79	5.65
Vitality at 72 h	28.92	12.89	15.38	9.76

## Data Availability

Data are contained within the article or Appendix A.

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
