# Peer review of "The Putative S1PR1 Modulator ACT-209905 Impairs Growth and Migration of Glioblastoma Cells In Vitro"

_cancers, 2023, doi:10.3390/cancers15174273_

Round 1
Reviewer 1 Report (Previous Reviewer 1)
My main concern is that unquestionable proof that the effects of ACT-209905 are mediated by S1PR1 is still lacking. In case authors are not in the position to do more experiments, my suggestion is to modify the title to emphasize the effects of ACT-209905 on glioma cells and reduce the emphasis on the receptor. In my opinion, the receptor shouldn’t be in the title, unless definitive proof is presented showing that the effects of ACT-209905 are indeed mediated by S1PR1. The quality of various figures should be improved to judge the results. In my opinion, those that cannot be improved should be replaced or removed in order to limit the message to what can be conclusively established.
I suggested a comparative analysis with FTY, which is a very well characterized modulator of S1PR1 and should have similar effects to those observed with ACT-209905. However, the authors preferred not to attend that suggestion. They now include a supplementary figure showing the effect of ACT-209905 on S1PR1 localization with an analysis done at 48 h. Which is very different to what would be expected from the characterized time course of S1PR1 internalization. For instance, FTY shows a clear effect at 30 min, see: (https://www.pnas.org/doi/10.1073/pnas.1014154108, Fig. 4). Since comparison with the the effect of FTY is missing, an alternative approach should be used to provide evidence that ACT-209905 effects are indeed mediated by S1PR1. An alternative is to test ACT-209905 effects in S1PR1 knockdown cells. I did not initially suggest knockdown because I understand that it might be more difficult. However, the lack of comparison with a well characterized effect (as FTY) makes necessary to provide alternative proof of the involvement of S1PR1. Since the weakest point in the manuscript is the lack of unquestionable proof that the effects of ACT-209905 are mediated by S1PR1, as I mentioned, in case the authors are not in the position to do more experiments, my suggestion is to modify the title to emphasize the effects of ACT-209905 on glioma cells and reduce the emphasis on the receptor. In my opinion, the receptor shouldn’t be in the title, unless definitive proof is presented showing that the effects of ACT-209905 are indeed mediated by S1PR1.
The quality of pictures showing the wound closure migration experiments is not good enough to judge the result. The result should be clearly visible without showing the arrows. From my point of view, as presented, these photographs are not suitable for publication. Since the Figures are still the same, I guess it is left to the editor to decide.
Perhaps there was a mistake in the preparation of the Figures included in the amended manuscript, but many labels are missing or need corrections. Protein names are covered by the graphs, many labels are absent in the pictures and graphs. Molecular weight markers are missing. For instance, the first version of Fig. 1 looks better. All the labels in the western blots of the new version are missing. The letters to identify some panels are also missing. Labels in the pictures and graphs of the new Fig. 2 are missing. The new Fig. 3 is missing most labels. The new Fig. 4 is missing most labels and the same situation occurs in the rest of the figures. It looks like a work in progress and not a definitive version.
Author Response
Please find our detailed response to the reviewers’ comments in the file enclosed below.

Reviewer 2 Report (Previous Reviewer 2)
Accept
Author Response
We thank the reviewer for the endorsement of our manuscript.
Reviewer 3 Report (Previous Reviewer 3)
In this study, the impact of the S1P receptor 1 (S1PR1) on growth and migration of glioblastoma (GBM) cells was explored. Viability and migration of GBM cells were reduced by the modulator of S1PR1 ACT-209905. Furthermore, co-culture with monocytic THP-1 cells or conditioned medium enhanced viability and migration of GBM cells suggesting that THP-1 cells secrete factors which stimulate GBM cell growth. ACT-209905 inhibited THP-1-induced enhancement of GBM cell growth and migration. Immunoblot analyses showed that ACT-209905 reduced activation of growth promoting kinases (p38, AKT1, ERK1/2) whereas THP-1 cells and conditioned medium caused an activation of these kinases. In addition, ACT-209905 diminished surface expression of pro-migratory molecules, and reduced CD62P positive GBM cells. In contrast, THP-1 cells increased ICAM-1 and P-Selectin content of GBM cells which was reversed by ACT-209905. This study provides support for the role of S1PR1 signaling in growth of GBM cells and gives a partial explanation for the pro-tumorigenic effects that macrophages have on GBM cells. The results demonstrate that S1PR1 expression is strongly up regulated in human glioblastoma and that there is an association between S1PR1 with patient´s survival times. The authors postulate that further preclinical and clinical evaluation of a pharmacological modulation of S1PR1 may provide a novel therapeutic principle in GBM although this is speculative at this point.
Author Response
We thank the reviewer for that positive comment.
Round 2
Reviewer 1 Report (Previous Reviewer 1)
As I mentioned in my previous review, there is no proof that the effects of ACT-209905 are mediated by S1PR1. Therefore, any implication that it does so is misleading. Since no proof is provided, I think the manuscript can be accepted if it removes from the title and abstract any implication that ACT-209905 is acting on S1PR1.
In my opinion, the current title:
“The sphingosine-1-phosphate receptor modulator ACT-209905 impairs growth and migration of Glioblastoma cells in vitro”
Should be modified to:
“ACT-209905 impairs growth and migration of Glioblastoma cells in vitro”
Some phrases in the simple summary should be removed or modified because they are implying findings not proven in the manuscript (or not consolidated in the scientific literature):
“… we report on the impact of the S1P receptor 1 (S1PR1) on growth and migration of glioblastoma (GBM) cells by using suitable in vitro models and pharmacological 30 modulation by ACT-209905”
Should be modified to:
“… we report on the impact of ACT-209905 on growth and migration of glioblastoma (GBM) cells by using suitable in vitro models and pharmacological modulation”.
“We now show that pharmacological modulation S1PR1 by ACT-209905 inhibits GBM cell growth and migration”
Should be modified to:
“We now show that ACT-209905 inhibits GBM cell growth and migration”
“We believe that this manuscript fits the interests of the readership of Cancers as it addresses S1PR1 as a promising target for improving GBM therapy”
Should be modified to:
“We believe that this manuscript fits the interests of the readership of Cancers as it addresses ACT-209905 as a promising molecule for improving GBM therapy”
In the abstract:
“Therefore, we investigated the role of S1PR1 in growth of human (primary cells, LN-18) and murine (GL261) GBM cells using S1PR1 modulator ACT-209905”
Should be modified to:
“Therefore, we investigated the effect of ACT-209905 (a putative S1PR1 modulator) in growth of human (primary cells, LN-18) and murine (GL261) GBM cells”
I also mentioned that, in my opinion, the quality of pictures showing the wound closure migration experiments is not good enough to judge the result. Since the Figures are still the same, I guess it is left to the Editor to decide.
Author Response
Please find our detailed response to the reviewer enclosed in the file below.

This manuscript is a resubmission of an earlier submission. The following is a list of the peer review reports and author responses from that submission.
Round 1
Reviewer 1 Report
Bien-Möller and colleagues study the effect of ACT-209905 on glioblastoma cell viability and migration.
Given the potential clinical relevance of S1PR1 expression in glioblastoma patients, authors investigate the effect of S1PR1 modulation on cellular processes linked to cancer progression. They also investigate the effect of ACT-209905 on the communication between THP-1 monocyte cells cocultured with Glioblastoma cells. Since there is no previous information on the the cellular and molecular effects of ACT-209905 (a search in PubMed finds zero articles), some basic analysis would strengthen the reported findings. For instance, what is the effect of ACT-209905 on S1PR1 expression and subcellular localization. Some comparative analysis with the effect of the well characterized functional antagonist FTY720 would be appropriate.
There are some mistakes in the text and figures, and some results are difficult to evaluate due to the quality of the pictures. Examples:
194: “On the day of invasion analysis” The method described corresponds to a chemotactic assay, not an invasion assay.
322: “In human LN-18 GBM cells, viability was significantly attenuated by 10, 20 and 50 322 μM ACT-209905 after 48h to 36.2%, .18.5% and 1.43%, respectively (Fig. 1d, upper panel).”, the Figure says “72h”.
325, 328, 333: “(Supplemental file S1a, upper panel)”, there is not supplemental information.
335: “In murine GL261 cells, already 1 μM ACT-209905 resulted in a significant loss of cell viability to 72% after 48h which was potentiated with increasing ACT-209905 concentration to a maximal reduction of 7.3% at 50 μM ACT-209905 (Fig. 1e, upper panel).”, The figure says 72h and the text says 48h.
Several results are indicated to be shown in supplementary figures, but the manuscript does not contain supplementary figures.
There are inconsistencies between the text and figures. For Fig. 1, the text and figure legends describe the effects at 48h, but the figure shows 72 h.
389: “…as standard chemotherapeutic for GBM 380 patients”, since patients are not studied, avoid referring to them in the subtitle.
The letters referring to each panel should be at the top left.
The quality of pictures showing the wound closure migration experiments is not good enough to judge the result. The result should be clearly visible without showing the arrows. It would help to fix and stain the cells at the end of the experiment.
Chemotactic experiments shown in Fig. 2f and 2g have marginal effects. The chemotactic effect of S1P is very limited, which makes difficult to interpret the effect of ACT-209905. The suggestion is to optimize or remove the migration experiments.
478: “Boyden chamber assays which reflect the invasion of cells through a collagene coated membrane”, chemotactic experiments shouldn’t be referred to as invasion experiments. This is not correct. Invasion is used in the case of proteolytic activity involved while cells migrate and secrete proteases to destroy a layer of extracellular matrix, which is not the case in the described experiments. Correct collagene.
Fig.5C and Fig.5D, upper panels, control conditions are experimentally similar, but in 5C there is almost no effect, although the graph shows a 50% increase on GL261 cell migration. Pictures that better represent the average results should be presented. Y axis should not be named “Invasion”, chemotactic effect would be more appropriate.
Fig. 6. Various results shown in the wb in Fig. 6 do not look consistent with the average effects shown in the graphs. Compare the 4th and 6th lanes which, according to the graphs, show inhibition. For instance, pERK in Fig. 6C.
The file that should show the original westerns does not contain them, it shows the same cropped westerns and figures included in the manuscript.
Were pERK and tERK westerns done in parallel or the same filters? The length of the bands in Fig. 6a and 6b look different. Some details on the figure preparation should be corrected. Some protein names are covered by the graphs. For instance, t-ERK in Fig. 6b.
The effect of chronic treatment with ACT on S1PR1 expression by wb and FACS should be assessed, in order to assess effects on the receptor itself.
Author Response
Please find our detailed response to the reviewer in the enclosed file “Response to Reviewer #1”.

Reviewer 2 Report
In this manuscript, the authors explore the impact of the SIPR1 antagonist on growth and migration of GBM cells, and also investigate the effect of co-culture of GBM cells with THP-1 cells on the GBM cells growth and migration. Finally, the authors also explored the potential relationship between signaling pathways and SIPR1 antagonists and THP-1 induced proliferation and migration of GBM cells. The authors present lots of experimental results. In general, my assessment of this manuscript is positive, however, my biggest concern is the SIPR1 antagonist, the antagonist may target many genes, not just the SIPR1 gene. So I suggest the authors can test the effect of SIPR1 knocking-down to GBM cells growth and migration. I think my biggest concern should be addressed before publication in Cancers.
Other specific comments:
1. In Figure1 a-b, I suggest the authors quantify the protein expression of these genes in different cell lines.
2. In figure 6, all the western blot quantity is not suitable for publication, please make a length to width ratio adjustment.
Author Response
Please find our detailed response to the reviewer in the enclosed file “Response to Reviewer #2”.

Reviewer 3 Report
In this study, the impact of the S1P receptor 1 (S1PR1) on growth and migration of glioblastoma (GBM) cells was explored. Viability and migration of GBM cells were reduced by the modulator of S1PR1 ACT-209905. Furthermore, co-culture with monocytic THP-1 cells or conditioned medium enhanced viability and migration of GBM cells suggesting that THP-1 cells secrete factors which stimulate GBM cell growth. ACT-209905 inhibited THP-1-induced enhancement of GBM cell growth and migration. Immunoblot analyses showed that ACT-209905 reduced activation of growth promoting kinases (p38, AKT1, ERK1/2) whereas THP-1 cells and conditioned medium caused an activation of these kinases. In addition, ACT-209905 diminished surface expression of pro-migratory molecules, and reduced CD62P positive GBM cells. In contrast, THP-1 cells increased ICAM-1 and P-Selectin content of GBM cells which was reversed by ACT-209905. This study provides support for the role of S1PR1 signaling in growth of GBM cells and gives a partial explanation for the pro-tumorigenic effects that macrophages have on GBM cells. The results demonstrate that S1PR1 expression is strongly up regulated in human glioblastoma and that there is an association between S1PR1 with patient´s survival times. The authors postulate that further preclinical and clinical evaluation of a pharmacological modulation of S1PR1 may provide a novel therapeutic principle in GBM although this is speculative at this point.
Minor English editing recommended.
Author Response
Please find our detailed response to the reviewer in the enclosed file “Response to Reviewer #3”.
